# From Past To Path: Masked History Learning for Next-Item Prediction in Generative Recommendation

## Abstract

Generative recommendation, which directly generates item identifiers, has emerged as a promising paradigm for recommendation systems. However, its potential is fundamentally constrained by the reliance on purely autoregressive training. This approach focuses solely on predicting the next item while ignoring the rich internal structure of a user's interaction history, thus failing to grasp the underlying intent. To address this limitation, we propose **Masked History Learning (MHL)**, a novel training framework that shifts the objective from simple next-step prediction to deep comprehension of history. MHL augments the standard autoregressive objective with an auxiliary task of reconstructing masked historical items, compelling the model to understand "why" an item path is formed from the user's past behaviors, rather than just "what" item comes next. We introduce two key contributions to enhance this framework: (1) an **entropy-guided masking** policy that intelligently targets the most informative historical items for reconstruction, and (2) a **curriculum learning** scheduler that progressively transitions from history reconstruction to future prediction. Experiments on three public datasets show that our method significantly outperforms state-of-the-art generative models, highlighting that a comprehensive understanding of the past is crucial for accurately predicting a user's future path. The code will be released to the public.

## 1 Introduction

Recommender systems have become essential tools for navigating the vast digital landscape, evolving from collaborative filtering (Wang et al., 2015; Li et al., 2024; Chen et al., 2018) to sequential models that capture user behavior dynamics (Purificato et al., 2024; Yuan et al., 2023; He et al., 2023). A new paradigm, *generative recommendation* (Rajput et al., 2023; Muennighoff et al., 2025), has recently emerged, offering powerful new ways to model user preferences. This approach adapts pre-trained language models like T5 (Rajput et al., 2023; Bao et al., 2023) and utilizes large language models (Hou et al., 2025b) to directly generate a sequence of semantic IDs representing the items to be recommended (Hua et al., 2023; Zhai et al., 2024), thus providing unprecedented flexibility.

However, despite their architectural diversity, these models share a fundamental limitation: they are trained almost exclusively to predict the next single item, rather than to understand the path that led there. This narrow focus on autoregressive *next-item prediction*, while intuitive, prioritizes local transitions over global understanding of user behavior. We argue that this paradigm produces models skilled at forecasting the immediate future but weak at understanding the user's **past**. It hinders the ability to capture crucial long-range dependencies and the underlying intent driving user behavior, limiting the accuracy to predict a complex user's **path** (see Appendix A for the pilot experiment).

For example, as shown in Fig. 1, consider the purchasing path of a photography enthusiast, who interacts with the following items in order: *camera body*, *tripod*, *camera bag*, and *camera lens*. Although the ground truth for the subsequent purchase is the *memory card*, existing models, fixated on recent item (*camera lens*), often incorrectly predict other lens-related accessories. The user's intention is a direct continuation of purchasing the initial *camera body*, but this intention is obscured by intermediate items. Due to the inability to fully internalize the underlying intention associations behind items along the purchasing path, existing autoregressive models are trained merely to predict "what comes next," but cannot effectively understand "why this path matters."

To address this limitation, we introduce **Masked History Learning** (MHL), a novel training framework for generative recommendation. Specifically, we augment next-item prediction with an auxiliary objective of reconstructing masked items within historical paths. This approach shifts the learning paradigm from predicting results to understanding the process, yielding three key advantages: **(a) Capturing Logical Dependencies.** MHL compels the model to understand the intrinsic associations between masked items and other items, thereby shifting the focus from statistical co-occurrence to the logical structure of a user's path. For example, by reconstructing the masked historical item "tripod", the model is forced to learn that "tripod" is a logical complement to a "camera body" purchase. **(b) Inferring Latent User Intent.** The deep understanding of paths enables models to look beyond a user's explicit behaviors and infer the latent intent driving them.

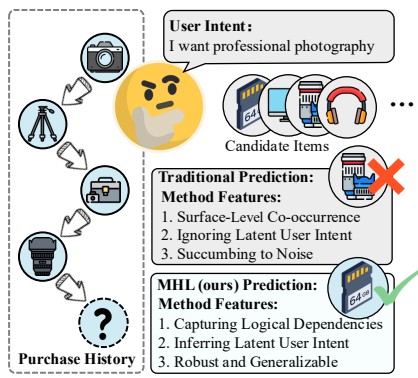

Figure 1: Prediction comparison between the traditional generative recommendation system and the proposed MHL framework.

Models can learn to comprehend a coherent and higher-level goal (such as an intent of "pursuing professional photography") from seemingly disparate items (such as cameras, bags, and future accessories). **(c) Learning Robust and Generalizable Representations.** The history reconstruction objective inherently enhances the quality of the learned representations. To reconstruct history accurately, the model must prioritize strong, logically consistent signals while learning to discount irrelevant or noisy interactions that provide poor contextual clues. This focus on the core signal results in item representations that are more stable and less susceptible to incidental deviations in behavior.

We validate the proposed MHL at multiple granularities: item-, token-, and mixed-level, consistently observing performance gains. To refine this learning process, we introduce two key innovations. First, moving beyond random masking, we propose an adaptive strategy guided by information theory (MacKay, 2002). We selectively mask items sharing high **entropy** with others, creating challenging training signals that focus on significant behavioral patterns. Second, we employ **curriculum learning** (Bengio et al., 2009) to connect the history reconstruction training with autoregressive inference. The training process begins with a warmup phase (He et al., 2016) using random masking, followed by a transition to a high masking ratio guided by entropy to build deep contextual understanding. The ratio is gradually reduced to prepare the model for path generation.

We conduct extensive experiments on three categories of the Amazon Reviews 2014 dataset (McAuley et al., 2015). The results demonstrate that understanding the past significantly enhances the model's ability to predict future paths, outperforming state-of-the-art baselines on metrics like Recall@K and NDCG@K. The contributions of this paper can be summarized as follows:

- We identify a key limitation in generative recommenders: training dominated by next-step prediction overlooks deep understanding of user history. We address this by proposing **Masked History Learning**, which jointly learns to reconstruct a user's *past* to better predict their future *path*.

- We design two strategies to enhance our framework: **entropy-guided masking** to focus on the most informative historical parts, and **curriculum learning** to bridge the gap between understanding history and generating future paths.

- Extensive experiments on three categories of the Amazon Reviews 2014 dataset validate our approach's effectiveness, achieving new state-of-the-art results for generative recommendation.

## 2 RELATED WORK

**Sequential Recommendation.** Sequential recommendation aims to model user behavior over time to predict future interactions. Early methods use Markov chains to capture item-to-item transitions (Rendle et al., 2010). Deep learning has since transformed this field. Modern approaches employ various neural architectures including recurrent neural networks (Hidasi et al., 2016; Li et al., 2017; Yue et al., 2024), convolutional neural networks (Tang & Wang, 2018), Transformers (Kang & McAuley, 2018; Sun et al., 2019), and graph neural networks (Chang et al., 2021; Wu et al., 2019).

While most sequential models are trained autoregressively, some studies have explored alternative learning objectives that go beyond simple next-item prediction. BERT4Rec (Sun et al., 2019) and S³-Rec (Zhou et al., 2020) use masked item prediction with bidirectional encoders to learn rich contextual representations for discriminative recommendation. These models randomly mask items in user sequences and learn to reconstruct them using full bidirectional context. This approach helps models capture richer dependencies compared to purely left-to-right training. Despite this success, the generative recommendation paradigm has remained largely autoregressive.

**Generative Recommendation.** Recent advances in generative models have introduced a new paradigm for recommendation systems. This approach shifts from discriminative to generative frameworks (Rajput et al., 2023). Inspired by generative retrieval (Tay et al., 2022; Wang et al., 2022), these methods tokenize items into discrete semantic identifiers. Sequence-to-sequence models can then directly generate these identifiers as recommendations.

Two main approaches have emerged in generative recommendation. The first leverages Large Language Models (LLMs) through zero-shot prompting (Gao et al., 2023; Harte et al., 2023) and instruction tuning (Muennighoff et al., 2025) to align LLMs with user behaviors. The second focuses on semantic ID-based generation, where items are first encoded as discrete token sequences (Rajput et al., 2023) derived from quantizing dense representations (Hua et al., 2023; Wang et al., 2024), then autoregressively decoded to produce recommendations (Zhai et al., 2024).

Despite their flexibility and scalability, existing generative recommenders (Rajput et al., 2023; Wang et al., 2024; Hou et al., 2025b) share a common limitation: they rely almost exclusively on autoregressive training that predicts the next item token given previous tokens. This left-to-right approach focuses on local transitions but may miss internal dependencies and underlying user intent.

**Our Contribution.** Our study addresses this gap by introducing history reconstruction learning to generative recommendation. Unlike BERT4Rec and S³-Rec, which employ masked prediction within bidirectional encoders to learn representations for discriminative scoring tasks, our proposed MHL augments standard *unidirectional, decoder-only* model with an auxiliary historical reconstruction objective. This design preserves the model's native autoregressive generation capability while enriching the training signal through deeper historical understanding. We further introduce entropy-guided masking to focus learning on the most informative historical patterns and curriculum learning to seamlessly transition from history understanding to future path generation. Together, these contributions establish a new training paradigm for generative recommenders that emphasizes understanding the past to better predict future paths.

## 3 METHOD

This section introduces proposed Masked History Learning (MHL) framework. MHL jointly learns to reconstruct a user's past and predict the future path. We enhance the framework with two strategies: entropy-guided masking and curriculum learning. We first present the preliminaries of generative recommendation. Then we detail the MHL framework and its two enhancement strategies.

### 3.1 PRELIMINARY

Generative recommendation models the recommendation task as an end-to-end sequence generation task. Given a sequence of user's historical interaction items $S_T = \{\phi(i_1), \ldots, \phi(i_T)\}$, each item $i_t \in \mathcal{I}$ is represented by its unique semantic ID (denoted as $w_{i_t}$):

$$\phi(i_t) = \{w_{i_t}^1, w_{i_t}^2, \ldots, w_{i_t}^K\} \tag{1}$$

which contains $K$ codewords, and each codeword at position $k$ belongs to a fixed codebook $\mathcal{W}^k$. The generative recommendation model is required to predict the semantic ID of the next item $\phi(i_{T+1})$.

Naturally, the training objective follows the standard autoregressive sequence generation task to maximize the conditional log-likelihood of the next item:

$$\max_\theta \log P_\theta\big(\phi(i_{T+1}) \mid S_T\big) = \max_\theta \sum_{k=1}^K \log P_\theta\big(w_{i_{T+1}}^k \mid S_T, w_{i_{T+1}}^{<k}\big) \tag{2}$$

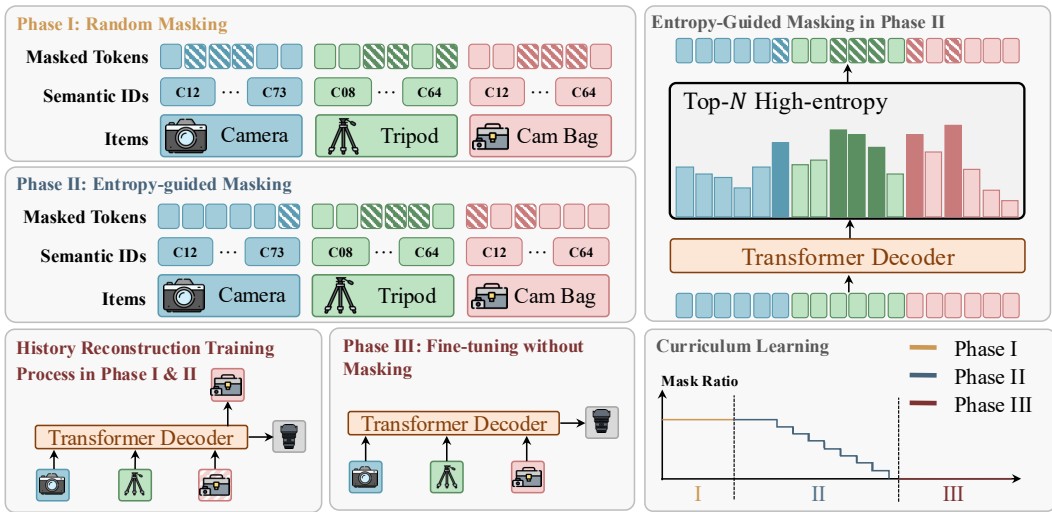

Figure 2: Overview of the proposed MHL framework. It enhances generative recommendation using a masked history learning objective. A curriculum training scheduler manages its three distinct phases, beginning with a random masking warm-up, transitioning to an adaptive entropy-guided masking strategy, and concluding with a fine-tuning stage without masking.

During inference, the model decodes the most probable semantic ID for a historical item sequence. The generated semantic ID is then used to retrieve the corresponding item from the catalog as the next predicted item for recommendation (Rajput et al., 2023).

## 3.2 MASKED HISTORY LEARNING FRAMEWORK

The above training objective of the conventional generative recommendation model mainly focuses on predicting the next item but neglects the learning of the ability to understand user history. Therefore, we propose MHL, which enables the model to reconstruct masked items in the user's historical sequence. Specifically, we have designed three masking strategies with different granularities.

**Item-level Masking.** This strategy masks entire items within the historical sequence. We select a subset of items and replace their whole semantic ID codewords with "[MASK]" tokens. For an selected item $i_t$, its item-level masked semantic ID representation is:

$$\tilde{\phi}(i_t) = \{[\text{MASK}^1], [\text{MASK}^2], \dots, [\text{MASK}^K]\} \tag{3}$$

It forces the model to reconstruct items from context and learn item-to-item dependencies.

**Token-level Masking.** This strategy masks individual codewords within the semantic ID sequence. For a selected subset of items, we replace one or more of their semantic ID codewords with "[MASK]" tokens. For selected item $i_t$, its token-level masked semantic ID can be:

$$\tilde{\phi}(i_t) = \{w_{i_t}^1, [\text{MASK}^2], w_{i_t}^3, \dots, [\text{MASK}^K]\} \tag{4}$$

It allows the model to learn fine-grained semantic relationships between codewords, which is crucial for modeling sub-item-level attributes and enhancing the generalization of item representation.

**Mixed Masking.** This strategy combines item- and token-level masking. For each selected item, we randomly choose between the two masking approaches to provide comprehensive training signals. It promotes a deeper understanding of both item-level and codeword-level relationships.

Base on the original historical sequence $S_T$, we apply the proposed mask strategies to obtain the masked sequence $\tilde{S}_T$ for training. We define a joint loss function with the following two objectives:

1) **Next-Item Prediction Loss**. It follows the popular autoregressive paradigm, predicting the semantic ID of the next item based on the masked historical sequence $\tilde{S}_T$. The loss is defined as:

$$\mathcal{L}_{\text{next}} = -\log P_\theta\left(\phi(i_{T+1}) \mid \tilde{S}_T\right) \tag{5}$$

In practice, it is the average of digit-wise cross-entropy over $K$ codeword positions.

2) **Masked History Reconstruction Loss**. It reconstructs original semantic IDs at masked positions in a non-autoregressive manner. Let $\mathcal{M}$ denote the set of all masked codewords in $\tilde{S}_T$. The reconstruction loss is defined as follows:

$$\mathcal{L}_{\text{mask}} = -\frac{1}{|\mathcal{M}|} \sum_{w \in \mathcal{M}} \log P_\theta\left(w \mid \tilde{S}_T\right) \tag{6}$$

It is noted that only the cross-entropy loss at masked positions is averaged.

The overall loss combines both next-item prediction and masked history reconstruction losses:

$$\mathcal{L}_{\text{MHL}} = \lambda_1 \cdot \mathcal{L}_{\text{next}} + \lambda_2 \cdot \mathcal{L}_{\text{mask}} \tag{7}$$

where $\lambda_1$ and $\lambda_2$ are weighting parameters to balance the two tasks. If there are no masked items, only the next-item prediction loss is retained.

## 3.3 Entropy-guided Masking

When we training with MHL framework, there is still a core challenge: which items should be selected for masking? An intuitive approach is to randomly select items for masking. It will serve as our baseline strategy. However, random masking may be inefficient for training, causing the model to easily predict frequent items and rendering reconstruction insignificant. More importantly, this fails to address our core motivation: understanding the logical dependencies and latent intent behind user paths. Therefore, we further introduce Entropy-Guided Masking into the proposed MHL framework, to alleviate both issues by intelligently masking the most challenging and informative positions in user history.

We measure prediction uncertainty using predictive entropy, where a high-entropy prediction indicates uncertain probability distributions, revealing that the model struggles to understand the reasons why the predicted item appears at its position in the interaction path. By precisely targeting and masking high-entropy positions, we implicitly force the model to reconstruct items via understanding underlying user intent and deeper contextual reasoning, thereby guiding the model to learn more robust and generalizable representations.

Specifically, for token-level masking, we compute entropy for each codeword in original user's historical sequence $S_T$. For any item $i_t$ in the path, its $K$ codewords are embedded (denoted as $\text{Emb}(\cdot)$) and then input into the mean-pooling layer (denoted as $\text{Mean-pooling}(\cdot)$) as the item representation:

$$E(i_t) = \text{Mean-pooling}\big(\text{Emb}(\phi(i_t))\big) \tag{8}$$

Then a transformer decoder (denoted as $\text{Dec}(\cdot)$) captures the dependencies among the representation sequences of items:

$$D_T = \text{Dec}(\{E(i_1), \ldots, E(i_T)\}) \tag{9}$$

For a codeword $w_{i_t}^k$ at position $k$ of item $i_t$, its entropy is:

$$H(w_{i_t}^k) = -\sum_{w \in \mathcal{W}^k} P_\theta^k\big(w \mid D_T\big) \log P_\theta^k\big(w \mid D_T\big), \tag{10}$$

where $\mathcal{W}^k$ is the codebook for the $k$-th position. The probabilities are computed with temperature scaling, $P_\theta^k(w \mid D_T) = \text{softmax}(\text{MLP}^k(D_T)/\tau)$.

For item-level masking, the entropy for an entire item $i_t$ is the aggregation of its constituent codeword entropies:

$$\bar{H}\big(\phi(i_t)\big) = \frac{1}{K} \sum_{k=1}^{K} H(w_{i_t}^k) \tag{11}$$

Based on the entropy scores, we can select top-$N$ tokens or items to mask.

### 3.4 CURRICULUM TRAINING SCHEDULER

While entropy-guided masking helps the model focus on understanding logical dependencies in user paths, our preliminary experiments indicate that a static masking strategy is usually sub-optimal. High-ratio masking from scratch may prevent the model from grasping basic sequential patterns before tackling complex user intent inference. In addition, the difference between the next-item prediction task and the masked history reconstruction task requires us to consider how to careful bridge the gap between historical understanding and future path generation in the learning process. Consequently, we design a curriculum training scheduler that gradually transitions from learning "why this path matters" to predicting "what comes next." The scheduler is divided into three phases.

**Phase I: Warm-up with Random Masking.** Training begins with low-complexity random masking. This allows the model to learn fundamental reconstruction and next-item prediction. It establishes a baseline understanding of sequential patterns.

**Phase II: Entropy-Guided Training with Adaptive Ratio.** After warm-up, we adopt entropy-guided masking to increase difficulty. We mask the top-$N$ high predicted entropy tokens/items, and set a masking ratio $\gamma$ to control the upper limit of $N$. For the token-level and item-level masking, $N$ is a random integer with a value range of $[1, \lfloor \gamma \cdot K \cdot T \rfloor]$ and $[1, \lfloor \gamma \cdot T \rfloor]$, respectively. Masking ratio adaptively decreases from an initial value of $\gamma_0$ as the validation performance plateaus.

**Phase III: Fine-tuning without Masking.** In the final phase, we set $\gamma = 0$ to remove the masked history reconstruction task. The model is trained on original historical sequence $S_T$ with only the next-item prediction objective. This ensures that the model can focus on fine-tuning next-item inference tasks. It mitigates train-test discrepancy and improves real-world performance.

## 4 EXPERIMENT

### 4.1 EXPERIMENTAL SETTINGS

**Dataset.** We evaluate models on three Amazon product categories: *Sports and Outdoors*, *Beauty*, and *Toys and Games* from the Amazon Reviews 2014 dataset (McAuley et al., 2015). We preprocess each category with core-5 filtering (He & McAuley, 2016). This retains only users and items with at least five interactions to ensure sufficient density for sequential modeling. For item metadata, we concatenate title, price, brand, feature, categories, and description into natural language sentences. This facilitates semantic representation learning following recent practice in generative recommendation (Wang et al., 2024). Table 10 from Appendix B shows detailed dataset statistics.

**Baselines.** We evaluate against comprehensive baselines in two categories: item ID-based methods and semantic ID-based approaches. Item ID-based methods operate directly on item IDs: GRU4Rec (Hidasi et al., 2016), HGN (Ma et al., 2019), SASRec (Kang & McAuley, 2018), FDSA (Hao et al., 2023), BERT4Rec (Sun et al., 2019), Caser (Tang & Wang, 2018), S$^3$-Rec (Zhou et al., 2020). Semantic ID-based approaches tokenize items into discrete semantic identifiers for generative recommendation: VQRec (Hou et al., 2023), RecJPQ (Petrov & Macdonald, 2024), TIGER (Rajput et al., 2023), HSTU (Zhai et al., 2024), RPG (Hou et al., 2025a).

**Evaluation Metrics.** We evaluate recommendation performance using Recall@K and NDCG@K with K=5 and 10. Following prior works (Kang & McAuley, 2018; Rajput et al., 2023; Sun et al., 2019; Hou et al., 2025a), we adopt standard leave-one-out strategy. For each user sequence, the last item is reserved for testing, the second-to-last for validation, and the remaining items for training.

**Implementation Details.** We encode item metadata (e.g., title, brand, price) with Sentence-T5-base (Ni et al., 2022). The resulting 768-dimensional embeddings are reduced to 128 dimensions via PCA and then discretized into sequences of 32 semantic tokens using FAISS-based optimized product quantization (OPQ) (Ge et al., 2014). Our backbone is a Transformer decoder, identical to the one used in RPG (Hou et al., 2025a), featuring a hidden size of 448, two layers, and four attention heads. The model is trained to jointly optimize next-item prediction and masked token reconstruction with equal weights. We apply an entropy-guided curriculum masking strategy, and early stopping is used when the mask ratio decays to zero. During optimization, we use AdamW with a learning rate of 5e-4, a batch size of 64, and cosine scheduling with 10k warmup steps. Inference is performed with graph-constrained beam search (Hou et al., 2025a) (beam size 50, 3

Table 1: Performance comparison of Item ID-based and Semantic ID-based models across three datasets. * denotes results reproduced using both the code and parameters from the authors.

| Model | Beauty | | | | Toys and Games | | | | Sports and Outdoors | | | |
|---|---|---|---|---|---|---|---|---|---|---|---|---|
| | R@5 | N@5 | R@10 | N@10 | R@5 | N@5 | R@10 | N@10 | R@5 | N@5 | R@10 | N@10 |
| **Item ID-based** | | | | | | | | | | | | |
| Caser | .0205 | .0131 | .0347 | .0176 | .0166 | .0107 | .0270 | .0141 | .0116 | .0072 | .0194 | .0097 |
| GRU4Rec | .0164 | .0099 | .0283 | .0137 | .0097 | .0059 | .0176 | .0084 | .0129 | .0086 | .0204 | .0110 |
| HGN | .0325 | .0206 | .0512 | .0266 | .0321 | .0221 | .0497 | .0277 | .0189 | .0120 | .0313 | .0159 |
| BERT4Rec | .0203 | .0124 | .0347 | .0170 | .0116 | .0071 | .0203 | .0099 | .0115 | .0075 | .0191 | .0099 |
| SASRec | .0387 | .0249 | .0605 | .0318 | .0463 | .0306 | .0675 | .0374 | .0233 | .0154 | .0350 | .0192 |
| FDSA | .0267 | .0163 | .0407 | .0208 | .0228 | .0140 | .0381 | .0189 | .0182 | .0122 | .0288 | .0156 |
| S3-Rec | .0387 | .0244 | .0647 | .0327 | .0443 | .0294 | .0700 | .0376 | .0251 | .0161 | .0385 | .0204 |
| **Semantic ID-based** | | | | | | | | | | | | |
| RecJPQ | .0311 | .0167 | .0482 | .0222 | .0331 | .0182 | .0484 | .0231 | .0141 | .0076 | .0220 | .0102 |
| VQ-Rec | .0457 | .0317 | .0664 | .0383 | .0497 | .0346 | .0737 | .0423 | .0208 | .0144 | .0300 | .0173 |
| TIGER | .0454 | .0321 | .0648 | .0384 | .0521 | .0371 | .0712 | .0432 | .0264 | .0181 | .0400 | .0225 |
| HSTU | .0469 | .0314 | .0704 | .0389 | .0433 | .0281 | .0669 | .0357 | .0258 | .0165 | .0414 | .0215 |
| RPG* | .0500 | .0358 | .0745 | .0436 | .0550 | .0386 | .0778 | .0460 | .0284 | .0197 | .0436 | .0246 |
| MHL (ours) | **.0574** | **.0424** | **.0795** | **.0495** | **.0656** | **.0471** | **.0885** | **.0544** | **.0342** | **.0243** | **.0484** | **.0289** |

propagation steps). The models are trained for up to 300 epochs on NVIDIA RTX A6000 GPUs. More details can be found in Appendix C.

## 4.2 EXPERIMENTS

**Overall Performance.** Table 1 presents the results across three Amazon product categories. We can find that MHL consistently achieves state-of-the-art performance. In addition, the results confirm that semantic ID-based models outperform traditional item ID-based approaches, with MHL leading all baselines, including strong competitors like TIGER and HSTU. The performance improvements are substantial. For example, MHL achieves a 27.1% improvement over TIGER in the NDCG@5 score for *Sports and Outdoors*. This validates our claim: understanding why a user path is formed is crucial for predicting what comes next. MHL's superior performance demonstrates three key benefits. First, by reconstructing masked historical items, the model learns logical dependencies between items rather than co-occurrence patterns. Second, the entropy-guided masking forces the model to focus on the most informative and challenging positions in user history, precisely where latent intent is obscured. Third, the curriculum learning bridges the gap between history understanding and future prediction, ensuring a smooth transition from learning "why this path matters" to predicting "what comes next". These targeted learning mechanisms enable MHL to consistently outperform baselines. The framework's effectiveness is particularly evident on the complex dataset like *Sports and Outdoors*, where logical item relationships are more nuanced and user intent is harder to infer.

**Ablation Study.** We conduct systematic ablation studies to understand each component's contribution within MHL. We evaluate six model variants across three masking strategies: Direct Inference (Inf) without masking, Random masking (R), and Entropy-guided masking (E). We also test three curriculum learning strategies: R→Inf, E→Inf, and the complete R→E→Inf framework. Table 2 validates our framework design through three key findings. First, all masking variants significantly outperform direct inference, demonstrating that reconstructing user history provides a richer learning signal. Second, entropy-guided masking consistently surpasses random masking, indicating that targeting high-entropy predictions is more effective for guiding the model to understand user intent. Finally, the complete R→E→Inf curriculum learning framework achieves optimal performance. This validates our curriculum design: starting with basic pattern learning through random masking, progressing to targeted understanding via entropy guidance, and finally fine-tuning for direct prediction. This progression mirrors the learning objective of transitioning from "why this path matters" to "what comes next".

## 4.3 FURTHER ANALYSIS

**Impact of Semantic ID Length.** We investigate how semantic ID length affects performance by varying codebook sizes from 4 to 64. As shown in Table 3, performance improves as codebook size increases from 4 to 32, then plateaus or slightly declines at 64. A codebook size of 32 consis-

Table 2: Ablation study comparing masking strategies and curriculum learning approaches with codebook size 32 and mask ratio 0.15.

| Mask Strategy | Curriculum Strategy | Beauty | | | | Toys and Games | | | | Sports and Outdoors | | | |
|---|---|---|---|---|---|---|---|---|---|---|---|---|---|
| | | R@5 | N@5 | R@10 | N@10 | R@5 | N@5 | R@10 | N@10 | R@5 | N@5 | R@10 | N@10 |
| No Mask | Direct Inference | .0482 | .0336 | .0704 | .0408 | .0485 | .0337 | .0713 | .0410 | .0267 | .0180 | .0393 | .0221 |
| Token-level | Random | .0547 | .0330 | **.0815** | .0417 | .0605 | .0356 | .0886 | .0446 | .0307 | .0173 | .0461 | .0222 |
| | Entropy-guided | .0538 | .0356 | .0802 | .0441 | .0601 | .0376 | **.0905** | .0474 | .0292 | .0182 | .0445 | .0231 |
| | R→Inf | .0533 | .0367 | .0793 | .0451 | .0609 | .0400 | .0895 | .0489 | **.0332** | **.0225** | .0478 | **.0272** |
| | E→Inf | .0535 | .0380 | .0774 | .0457 | .0570 | .0388 | .0838 | .0474 | .0224 | .0149 | .0344 | .0188 |
| | R→E→Inf | **.0568** | **.0411** | .0814 | **.0490** | **.0634** | **.0458** | .0880 | **.0538** | .0191 | .0123 | .0296 | .0156 |
| Item-level | Random | .0482 | .0340 | .0706 | .0413 | .0482 | .0331 | .0720 | .0407 | .0269 | .0183 | .0409 | .0228 |
| | Entropy-guided | .0491 | .0338 | .0726 | .0414 | .0490 | .0337 | .0719 | .0410 | .0251 | .0171 | .0372 | .0210 |
| | R→Inf | .0491 | .0343 | .0718 | .0416 | .0496 | .0342 | .0747 | .0423 | .0274 | .0186 | .0419 | .0232 |
| | E→Inf | .0516 | .0359 | **.0748** | .0433 | .0513 | .0358 | .0749 | .0433 | .0266 | .0182 | .0400 | .0225 |
| | R→E→Inf | **.0523** | **.0368** | .0743 | **.0439** | **.0553** | **.0384** | **.0755** | **.0449** | **.0286** | **.0193** | **.0421** | **.0236** |
| Mixed-level | Random | .0521 | .0350 | .0805 | .0441 | .0539 | .0347 | .0825 | .0440 | .0321 | .0195 | .0481 | .0247 |
| | Entropy-guided | .0524 | .0352 | **.0806** | .0443 | .0561 | .0367 | .0854 | .0462 | .0323 | .0210 | **.0492** | .0264 |
| | R→Inf | .0535 | .0374 | .0803 | .0461 | .0556 | .0384 | .0815 | .0468 | **.0329** | .0218 | .0490 | .0269 |
| | E→Inf | .0530 | .0372 | .0785 | .0454 | .0595 | .0398 | **.0869** | .0486 | .0267 | .0180 | .0396 | .0221 |
| | R→E→Inf | **.0537** | **.0383** | .0784 | **.0463** | **.0613** | **.0434** | .0855 | **.0511** | .0327 | **.0223** | .0484 | **.0274** |

tently achieves optimal performance across datasets, particularly for the mixed-level. Smaller codebooks (size 4) lack the capacity to capture rich semantic information, yielding suboptimal results. Conversely, larger codebooks (size 64) may introduce sparsity or excessive complexity, leading to marginal performance decline. These findings suggest that semantic ID length must balance expressiveness with learnability. Size 32 provides an optimal trade-off, enabling diverse and meaningful semantic representations without overwhelming the model with unnecessary complexity.

Table 3: Performance comparison across different codebook sizes with mask ratio 0.15.

| Mask Strategy | Codebook Size | Beauty | | | | Toys and Games | | | | Sports and Outdoors | | | |
|---|---|---|---|---|---|---|---|---|---|---|---|---|---|
| | | R@5 | N@5 | R@10 | N@10 | R@5 | N@5 | R@10 | N@10 | R@5 | N@5 | R@10 | N@10 |
| Token-level | 4 | .0406 | .0291 | .0568 | .0343 | .0434 | .0298 | .0638 | .0363 | .0179 | .0124 | .0273 | .0154 |
| | 8 | .0502 | .0369 | .0691 | .0430 | .0577 | .0409 | .0805 | .0482 | .0280 | .0209 | .0395 | .0245 |
| | 16 | **.0574** | **.0424** | .0795 | **.0495** | **.0644** | .0456 | **.0883** | .0533 | **.0334** | **.0239** | .0474 | **.0284** |
| | 32 | .0568 | .0411 | .0814 | .0490 | .0634 | **.0458** | .0880 | **.0538** | .0191 | .0123 | .0296 | .0156 |
| | 64 | .0552 | .0390 | .0805 | .0472 | .0456 | .0318 | .0659 | .0383 | .0325 | .0226 | **.0475** | .0275 |
| Item-level | 4 | .0344 | .0253 | .0516 | .0308 | .0385 | .0259 | .0576 | .0320 | .0151 | .0107 | .0242 | .0136 |
| | 8 | .0458 | .0329 | .0646 | .0390 | .0503 | .0359 | .0716 | .0427 | .0249 | .0180 | .0366 | .0218 |
| | 16 | .0501 | .0363 | .0704 | .0429 | **.0553** | **.0390** | **.0802** | **.0470** | .0265 | .0184 | .0403 | .0227 |
| | 32 | **.0523** | **.0368** | **.0743** | **.0439** | **.0553** | .0384 | .0755 | .0449 | **.0286** | **.0193** | **.0421** | **.0236** |
| | 64 | .0509 | .0358 | .0731 | .0429 | .0494 | .0347 | .0678 | .0406 | .0273 | .0179 | .0417 | .0226 |
| Mixed-level | 4 | .0376 | .0272 | .0531 | .0322 | .0422 | .0287 | .0603 | .0346 | .0203 | .0146 | .0302 | .0178 |
| | 8 | .0468 | .0343 | .0645 | .0400 | .0553 | .0385 | .0785 | .0460 | .0279 | .0201 | .0389 | .0237 |
| | 16 | **.0537** | **.0384** | .0757 | .0455 | .0592 | .0421 | .0845 | .0503 | .0307 | .0212 | .0447 | .0257 |
| | 32 | **.0537** | .0383 | **.0784** | **.0463** | **.0613** | **.0434** | **.0855** | **.0511** | **.0327** | **.0223** | **.0484** | **.0274** |
| | 64 | .0533 | .0377 | .0760 | .0451 | .0579 | .0411 | .0824 | .0490 | .0318 | .0218 | .0471 | .0267 |

**Sensitivity to Masking Ratio.** Table 4 examines how different masking ratios (0.05 to 0.40) affect performance when the Beauty dataset uses a codebook size of 32, while Toys and Games and Sports and Outdoors use a 16-bit codebook. Results show that MHL is relatively stable across ratios, with slight variations depending on the masking strategy and dataset. For token-level masking, ratios between 0.10 and 0.25 yield consistently strong performance. Item-level masking shows similar robustness, with 0.15 often achieving optimal results. Mixed-level masking maintains reasonable performance across ratios but does not consistently outperform the specialized strategies, suggesting that the combination approach provides a middle ground rather than universal superiority. This stability indicates that MHL is robust to mask ratios, making it practical for real-world deployment.

**Impact of Reconstruction Loss Weight.** To evaluate the role of the reconstruction objective in MHL, we vary the reconstruction loss weight from 0.2 to 1.0 while keeping the prediction loss fixed at 1.0. Table 5 reports token-level performance on Beauty, Toys and Games, and Sports and Outdoors datasets. The results show that higher reconstruction weights generally lead to better

Table 4: Performance Sensitivity Analysis across Different Mask Ratios with Dataset-Specific Codebook Sizes (32 for Beauty, 16 for Toys and Games / Sports and Outdoors).

| Mask Strategy | Mask Ratio | Beauty | | | | Toys and Games | | | | Sports and Outdoors | | | |
|---|---|---|---|---|---|---|---|---|---|---|---|---|---|
| | | R@5 | N@5 | R@10 | N@10 | R@5 | N@5 | R@10 | N@10 | R@5 | N@5 | R@10 | N@10 |
| Token-level | 0.05 | **.0571** | .0408 | .0811 | .0485 | .0522 | .0357 | .0730 | .0424 | .0245 | .0170 | .0365 | .0209 |
| | 0.10 | .0567 | .0404 | **.0817** | .0484 | **.0656** | **.0471** | **.0885** | **.0544** | .0139 | .0093 | .0230 | .0122 |
| | 0.15 | .0568 | **.0411** | .0814 | **.0490** | .0644 | .0456 | .0883 | .0533 | .0334 | .0239 | .0474 | .0284 |
| | 0.20 | .0550 | .0398 | .0781 | .0472 | .0610 | .0437 | .0861 | .0518 | .0342 | .0243 | .0484 | **.0289** |
| | 0.25 | .0562 | .0402 | .0803 | .0479 | .0611 | .0436 | .0870 | .0519 | **.0347** | **.0244** | .0474 | .0285 |
| | 0.30 | .0554 | .0397 | .0796 | .0475 | .0623 | .0442 | .0874 | .0522 | .0339 | .0235 | .0478 | .0280 |
| | 0.35 | .0558 | .0394 | .0791 | .0470 | .0603 | .0422 | .0858 | .0504 | .0341 | .0241 | **.0485** | .0287 |
| | 0.40 | .0546 | .0389 | .0785 | .0466 | .0596 | .0421 | .0851 | .0503 | .0325 | .0236 | .0452 | .0276 |
| Item-level | 0.05 | .0353 | .0242 | .0551 | .0306 | .0549 | .0383 | .0793 | .0462 | .0268 | .0185 | .0396 | .0227 |
| | 0.10 | .0519 | .0366 | .0742 | .0438 | .0582 | **.0417** | .0812 | **.0491** | .0265 | .0181 | .0392 | .0221 |
| | 0.15 | .0523 | .0368 | .0743 | .0439 | .0553 | .0390 | .0802 | .0470 | .0265 | .0184 | .0403 | .0227 |
| | 0.20 | .0517 | .0368 | .0734 | .0438 | .0563 | .0402 | .0803 | .0479 | .0274 | .0188 | .0399 | .0228 |
| | 0.25 | .0519 | .0371 | .0740 | .0442 | .0581 | .0407 | **.0834** | .0488 | .0267 | .0181 | .0401 | .0224 |
| | 0.30 | **.0541** | **.0379** | **.0788** | **.0458** | **.0584** | .0411 | .0820 | .0487 | .0242 | .0167 | .0369 | .0207 |
| | 0.35 | .0528 | .0376 | .0769 | .0453 | .0573 | .0400 | .0821 | .0479 | **.0283** | **.0198** | **.0420** | **.0242** |
| | 0.40 | .0505 | .0358 | .0750 | .0437 | .0563 | .0395 | .0800 | .0471 | .0252 | .0169 | .0374 | .0208 |
| Mixed-level | 0.05 | .0525 | .0373 | .0766 | .0450 | .0571 | .0406 | .0856 | .0497 | .0316 | .0222 | .0462 | .0269 |
| | 0.10 | .0539 | .0378 | .0782 | .0456 | .0599 | .0423 | .0834 | .0499 | **.0334** | **.0232** | .0469 | **.0276** |
| | 0.15 | .0537 | **.0383** | .0784 | .0463 | .0592 | .0421 | .0845 | .0503 | .0307 | .0212 | .0447 | .0257 |
| | 0.20 | .0537 | .0380 | .0781 | .0459 | .0580 | .0414 | .0835 | .0496 | .0331 | .0230 | .0468 | .0274 |
| | 0.25 | **.0544** | **.0383** | **.0797** | **.0464** | .0609 | **.0426** | .0858 | **.0507** | .0300 | .0206 | .0451 | .0255 |
| | 0.30 | .0529 | .0372 | .0767 | .0449 | **.0610** | .0422 | **.0873** | .0506 | .0317 | .0223 | .0468 | .0271 |
| | 0.35 | .0530 | .0375 | .0776 | .0454 | .0591 | .0418 | .0840 | .0498 | .0313 | .0222 | .0454 | .0267 |
| | 0.40 | .0524 | .0369 | .0759 | .0445 | .0568 | .0395 | .0825 | .0478 | .0315 | .0219 | **.0470** | .0269 |

Table 5: Token-level performance under varying reconstruction loss values (predict loss fixed at 1.0). Mask ratios: 0.15 for Beauty, 0.10 for Toys, 0.20 for Sports; codebook sizes: 32 for Beauty and 16 for Toys/Sports.

| Mask Strategy | Reconstruction Loss | Beauty | | | | Toys and Games | | | | Sports and Outdoors | | | |
|---|---|---|---|---|---|---|---|---|---|---|---|---|---|
| | | R@5 | N@5 | R@10 | N@10 | R@5 | N@5 | R@10 | N@10 | R@5 | N@5 | R@10 | N@10 |
| Token-level | 1.0 | **.0574** | **.0424** | **.0795** | **.0495** | **.0656** | **.0471** | **.0885** | **.0544** | .0342 | .0243 | **.0484** | .0289 |
| | 0.8 | .0554 | .0404 | .0793 | .0481 | .0628 | .0454 | .0865 | .0530 | **.0342** | **.0246** | .0480 | **.0290** |
| | 0.6 | .0549 | .0401 | .0790 | .0479 | .0614 | .0438 | .0856 | .0515 | .0314 | .0222 | .0456 | .0269 |
| | 0.4 | .0533 | .0382 | .0768 | .0458 | .0593 | .0423 | .0849 | .0506 | .0321 | .0225 | .0462 | .0270 |
| | 0.2 | .0526 | .0376 | .0758 | .0450 | .0587 | .0413 | .0834 | .0493 | .0327 | .0226 | .0461 | .0269 |

recall and NDCG scores, with the best performance achieved at a weight of 1.0, indicating that the reconstruction task provides effective self-supervised signals that complement the prediction loss. Performance remains relatively stable in the 0.8–1.0 range but gradually declines when the weight drops below 0.8, highlighting the importance of the reconstruction objective in learning meaningful token representations. This suggests that improvements are not solely due to the prediction loss: the masked reconstruction task itself significantly contributes to model performance, validating the design choice of masking and reconstructing tokens during training and confirming the robustness of MHL.

**Generalizability to Text Sequences.** To demonstrate MHL's broader applicability, we apply our framework to unstructured token sequences derived directly from item titles. Table 6 compares our complete R→E→Inf strategy against RPG baseline on raw text tokens. MHL consistently outperforms RPG across all metrics on the Beauty dataset. This result is significant because it shows that MHL's core principle, reconstructing the past to predict the future, generalizes beyond discrete semantic IDs to complex and noisy text sequences. The success on text sequences validates that MHL captures fundamental learning dynamics rather than exploiting specific properties of semantic ID representations. This generalizability highlights MHL's potential for broader applications in sequential modeling where understanding historical context is crucial for future prediction.

**Case Study.** As illustrated in Table 7, the baseline RPG model, trained solely on autoregressive next-item prediction, often misinterprets a user's intent by overemphasizing transient, noisy signals, such as the mid-sequence clothing items. For example, its predictions for items like "Crew Sock"

Table 6: Generalization study comparing MHL with RPG baseline on text-based token sequences using mixed masking strategy.

| Mask Strategy | Training Method | Beauty | | | | Toys and Games | | | | Sports and Outdoors | | | |
|---|---|---|---|---|---|---|---|---|---|---|---|---|---|
| | | R@5 | N@5 | R@10 | N@10 | R@5 | N@5 | R@10 | N@10 | R@5 | N@5 | R@10 | N@10 |
| Text w/o Mask | RPG | .0297 | .0212 | .0439 | .0258 | .0323 | .0234 | .0446 | .0273 | .0134 | .0094 | .0203 | .0117 |
| MHL (ours) | R→E→Inf | .0338 | .0238 | .0483 | .0285 | .0347 | .0249 | .0498 | .0297 | .0150 | .0106 | .0237 | .0134 |

deviate from the user's primary and recurring interest in athletic gear and accessories. In contrast, the proposed MHL framework, by requiring the model to reconstruct a user's historical trajectory, encourages it to identify and prioritize the core underlying intent. As a result, MHL can look beyond short-term deviations and accurately predict the next item "Youth Multi-Sport Helmet", which aligns logically with the user's sustained interest in firearm-related products.

Table 7: Case study comparison between MHL and the RPG baseline.

| Historical Purchase Sequence | |
|---|---|
| Footwear Adhesive → Running Waist Pack → Cardio Trampoline → Heavyweight T-Shirt → BMX Pads → ? | |
| **MHL Prediction (Top-5)** | **RPG Prediction (Top-5)** |
| Youth Multi-Sport Helmet ✓ | Crew Sock |
| NBA Street Basketball | Eco Open Bottom Pant |
| Mini Basketball Hoop | Training T-shirt |
| Indoor/Outdoor Basketball | Jersey Pants |
| NBA Game Ball Mini | Long Sleeve Cotton T-Shirt |

## 5 CONCLUSION

Existing generative recommenders focus on predicting "what comes next" but fail to understand "why this path matters". We introduce MHL, a simple and effective framework that learns from masked history reconstruction alongside next-item prediction. MHL incorporates entropy-guided masking to target informative historical positions and curriculum learning to transition from history understanding to future prediction. Experiments on three datasets show state-of-the-art performance and successful generalization to text-based Item IDs. Our findings confirm that understanding the past is crucial for predicting the future. MHL represents a significant step toward recommendation systems that comprehend user behavior patterns rather than merely statistical co-occurrence.

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

## A    PILOT EXPERIMENT

To illustrate the motivation of this paper, we conduct a pilot experiment to examine whether models trained with the standard next-item prediction paradigm overly rely on recent interactions, potentially neglecting the user's past behaviors. Specifically, on the *Toys and Games* dataset, for sequences longer than 20 in the test set, we remove the last 15 items and treat the truncated sequences as a new test set, using the first item of each truncated sequence as the prediction target to evaluate the models' ability to capture both short-term and long-term dependencies.

The results of full-sequence and truncated-sequence evaluation are shown in Table 8. Under the full sequence setting, MHL outperforms RPG across all metrics, achieving an 18.23% improvement on N@10. In the truncated setting, which emphasizes longer-range dependencies, the improvement is even larger, reaching 43.95%, indicating that MHL not only captures both short-term and long-term user preferences, but also better understands the overall sequence context. This comparison demonstrates that MHL more effectively models user behavior, whereas RPG tends to rely more heavily on recent interactions.

To further validate MHL's ability to capture long-term user intent, we conduct a length-stratified analysis, reported in Table 9. We bucket the test sequences by length and compute N@10 for both RPG and MHL. RPG exhibits an inverted-U performance curve: it performs reasonably well on medium-length sequences but struggles on very short or very long sequences. For instance, sequences longer than 50 items see RPG's N@10 drop to 0.0375, while MHL boosts it to 0.0577, corresponding to a 53.33% relative improvement. Overall, MHL consistently outperforms RPG across all length buckets, and the relative improvement is most pronounced for the extremely long sequences. These results confirm that MHL effectively captures long-term user preferences rather than relying primarily on recent interactions, further supporting the motivation for our proposed approach.

## B    STATISTICS OF THE DATASET

The detailed statistics of Amazon Reviews 2014 datasets is shown in Table 10.

## C    DETAILED IMPLEMENTAL DETAILS

We encode item metadata (title, brand, price, features, categories, description) using Sentence-T5 and reduce 768-dimensional embeddings to 128 dimensions with PCA. Following RPG (Hou et al., 2025a), we discretize continuous representations into generative semantic IDs using FAISS-based OPQ. Each item is represented as a sequence of 32 tokens (32 codebooks with 256 codewords each).

Table 8: Comparison of RPG and MHL on the *Toys* dataset. "Full" denotes evaluation on complete sequences, while "Truncated" denotes evaluation on the prefixes of long sequences. The last column reports relative N@10 improvement from Full to Truncated.

| Setting | Model | Toys and Games | | | | N@10 |
|---|---|---|---|---|---|---|
| | | R@5 | N@5 | R@10 | N@10 | ↑ (%) |
| Full | RPG | .0550 | .0386 | .0778 | .0460 | – |
| Full | MHL (ours) | .0656 | .0471 | .0885 | .0544 | 18.23% |
| Truncated | RPG | .6355 | .4823 | .7454 | .5176 | – |
| Truncated | MHL (ours) | .8460 | .7208 | .9199 | .7451 | 43.95 % |

Table 9: Length-stratified N@10 performance of RPG and MHL on the *Toys and Games* test set. The last column shows the relative improvement of MHL over RPG.

| Test Set | Toys and Games | | Rel. Improvement (%) |
|---|---|---|---|
| | **RPG N@10** | **MHL N@10** | |
| Full | 0.0460 | 0.0544 | 18.23 |
| ≤ 10 | 0.0475 | 0.0548 | 15.37 |
| (10,20] | 0.0377 | 0.0515 | 36.61 |
| (20,30] | 0.0332 | 0.0426 | 28.31 |
| (30,40] | 0.0621 | 0.0680 | 9.49 |
| (40,50] | 0.0653 | 0.0912 | 39.60 |
| > 50 | 0.0375 | 0.0577 | 53.33 |

Our backbone is a Transformer decoder with the same parameter size as RPG (Hou et al., 2025a): hidden size 448, 2 layers, 4 attention heads, feed-forward dimension 1024, and GELU activation. Maximum sequence length is 50 with dropout 0.3 for embeddings and attention modules.

For training, we jointly optimize next-item prediction and masked token reconstruction with equal weights. We use entropy-guided curriculum masking: training starts with random masking, then switches to entropy-based masking; if validation does not improve for 5 consecutive evaluations, mask ratio decays linearly by $0.1 \times r_0$ (with $r_0 = 0.15$) until reaching 0. After this, the model trains purely on prediction with early stopping patience of 20. Entropy forward propagation stabilizes masking decisions using window size 3, decay factor 2.0, and residual mixing coefficient 0.2 across item-level and token-level entropies.

During inference, we follow RPG (Hou et al., 2025a) and apply graph-constrained beam search with beam size 50, each node keeping 50 edges, and 3 propagation steps. Optimization uses AdamW with learning rate 5e-4, batch size 64, weight decay 0.0, gradient clipping 1.0, 10k warmup steps, and cosine learning rate scheduling. We train for up to 300 epochs with early stopping patience of 20. All experiments use NVIDIA RTX A6000 GPUs with distributed training and mixed precision.

Table 10: Statistics of the Amazon Reviews 2014 datasets. "Avg. $t$" denotes the average number of interactions per input sequence.

| Datasets | #Users | #Items | #Interactions | Avg. $t$ |
|---|---|---|---|---|
| Beauty | 22,363 | 12,101 | 176,139 | 8.87 |
| Toys and Games | 19,412 | 11,924 | 148,185 | 8.63 |
| Sports and Outdoors | 18,357 | 35,598 | 260,739 | 8.32 |

