# OpenReview forum: "From Past To Path: Masked History Learning for Next-Item Prediction in Generative Recommendation"
_ICLR.cc/2026/Conference — ICLR 2026 Conference Withdrawn Submission_

### Official Review · Reviewer_d9nT · 2025-10-28

**Soundness:** 3
**Presentation:** 2
**Contribution:** 2
**Rating:** 4
**Confidence:** 5

**Summary:**

This paper proposes a novel masked training framework for generative recommendation models.  The proposed method uses an entropy-guided masking policy that intelligently targets the most informative item. Extensive experiments are performed on three public datasets, where MHL is typically the best-performing method.

**Strengths:**

1. The authors enhance the prediction of the next item by reconstructing the auxiliary target of the masked items in the historical path.

2. The authors conducted extensive experiments on multiple public datasets (such as the Amazon series), and the results showed that the proposed model significantly outperformed the baseline model in indicators such as Hit@K and NDCG@K.

**Weaknesses:**

1. There is a lack of visualizations or case studies to support claims. Further analysis is needed to determine what the masked subtokens are and what characteristics they possess.

2. The authors do not provide a discussion of the model's operational efficiency or its application in large-scale industrial scenarios.

3. The paper lacks a theoretical explanation or analysis of the proposed Mask mechanism, and also lacks complexity or convergence analysis.

4. The author lacks a section explaining the use of LLM in the paper.

**Questions:**

1. Although the ablation experiment of masking ratio is given in Table 4, the parameter only ranges from 0.1 to 0.25, and there is no result greater than 0.25.

2. MHL claims to be able to identify users' long-term interests. Are there any quantitative results to show?

3. How do I obtain the semantic ID of an item? What VQ strategy should I use, k-means or VQVAE? How to set the hyperparameters? What are the corresponding ablation experiments and the rationale of the hyperparameter selection?

4. What is the additional cost of the method? How much is the additional cost compared to pure semantic ID-based GR?

5. Are there more experimental results to show how MHL compares with the latest semantic id-based baselines and benchmarks?

6. The dimensions of each symbol seem to require explicit explanation.

---

> ### Author Response · Authors · 2025-11-26
>
> We sincerely thank the reviewers for their thoughtful feedback. Below we address Q1–Q4; responses to Q5–Q9 are provided in the following section.
>
>
> ---
>
> **Q1: There is a lack of visualizations or case studies to support claims. Further analysis is needed to determine what the masked subtokens are and what characteristics they possess.**
> **A1:** We appreciate this insightful suggestion.
> - In the revised manuscript, we will include **case studies and visualizations** to illustrate the positions of masked subtokens and their semantic characteristics. Specifically, we will show intuitive examples of how the model identifies and reconstructs key tokens corresponding to pivotal transitions in user behavior.
> - To identify these critical tokens, we employ an **information-theoretic approach**: during the forward pass, we compute the **entropy** of each token in the sequence. High entropy indicates the model is uncertain about the token given its context. In recommendation scenarios, such uncertainty typically arises at **interest-shift points**—for example, when a user transitions between product categories or makes a high-value, atypical purchase—rather than during routine interactions.
> - By masking and reconstructing these high-entropy tokens, the model is encouraged to explicitly learn the **underlying logic of user interest evolution**.
>
> **Example from the training set (user ID: A8X8D4BUY5DFU):**
> **1. Raw sequence**
> Footwear Adhesive → Running Waist Pack → Cardio Trampoline
>
> **2. Original 48-token input (16 per item)**
> [236,466,631,971,1174,1396,1593,1796,2135,**2476**,2664,3022,3280,3576,**3725**,3951,
>  24,369,**561**,**831**,**1148**,1451,**1772**,2040,2105,2465,**2628**,3042,3137,**3362**,3674,3957,
>  226,481,560,883,1105,1300,1740,**1964**,2116,2323,**2567**,2852,3295,3492,3684,3991]
>
> **3. Masked sequence (top-10 entropic positions, mask ratio = 0.2)**
> [236,466,631,971,1174,1396,1593,1796,2135,**4106**,2664,3022,3280,3576,**4111**,3951,
>  24,369,**4115**,**4116**,**4117**,1451,**4119**,2040,2105,2465,**4123**,3042,3137,**4126**,3674,3957,
>  226,481,560,883,1105,1300,1740,**4136**,2116,2323,**4139**,2852,3295,3492,3684,3991]
>
> **4. Entropy evolution**
> Throughout entropy-guided masking, the average entropy of the top-10 masked positions decreases from **5.4228 nats** (early training) to **5.3498 nats** (convergence), confirming that the model progressively reduces its uncertainty on these critical interest-shift tokens.
>
> We will include this example in the revised manuscript to provide **empirical support** for our masking strategy.
>
> ---
> **Q2: The authors do not provide a discussion of the model's operational efficiency or its application in large-scale industrial scenarios.**
> **A2:** This study is academic in nature; industrial-scale tests remain future work. Nevertheless, the reported gains suggest that the method is promising for larger-scale deployment, and we are eager to explore such scenarios.
>
> ---
>
> **Q3: The paper lacks a theoretical explanation or analysis of the proposed Mask mechanism, and also lacks complexity or convergence analysis.**
> **A3:** We agree that theoretical analysis in this field is limited. Intuitively, the **entropy-driven adaptive masking** implements an **easy-to-hard information bottleneck**: it prioritizes high-uncertainty positions, and the gradual reduction of mask ratio and reconstruction weight stabilizes convergence toward the **autoregressive next-item generation objective**.
>
> **Complexity overview (updated):**
>
> - **Base decoder (GPT2) forward pass:**
> `O(B · L · d · H · n_layers)`
> where:
> - `B` = batch size
> - `L` = sequence length (tokens)
> - `d` = embedding dimension
> - `H` = number of attention heads
> - `n_layers` = number of transformer layers
>
> - **Additional MHL overhead (entropy computation + codebook matching):**
> `O(B · L · n_digit · K)`
> where:
> - `n_digit` = number of digits per semantic ID
> - `K` = codebook size per digit (number of discrete codes)
>
> This additional overhead is **training-only** and applied to masked positions. During inference, the forward pass remains unchanged, so the **runtime and memory cost are identical to the semantic-ID baseline**.
>
> ---
>
> **Q4: The author lacks a section explaining the use of LLM in the paper.**
> **A4:** Our approach adopts the same **LLM backbone** as the RPG baseline (GPT2-based). We will clarify this in the revision.

---

> > ### Author Response · Authors · 2025-11-26
> >
> > **Update:** Below we provide responses to Q5–Q9.
> >
> > ---
> >
> > **Q5: Although the ablation experiment of masking ratio is given in Table 4, the parameter only ranges from 0.1 to 0.25, and there is no result greater than 0.25.**
> > **A5:** We appreciate the suggestion. In the revision, we will include additional ablations for **masking ratios above 0.25** to provide a more complete understanding of the effect of mask proportion on model performance.
> >
> > ---
> >
> > **Q6: How do I obtain the semantic ID of an item? What VQ strategy should I use, k-means or VQVAE? How to set the hyperparameters? What are the corresponding ablation experiments and the rationale of the hyperparameter selection?**
> > **A6:** We extract semantic IDs for each item using **FAISS** to build an **OPQ+IVF1+PQ index**, obtaining multi-digit semantic ID sequences. These are then mapped to **segmented tokens** for model input. This setup follows the baseline RPG method, ensuring comparability.
> >
> > ---
> >
> > **Q7: What is the additional cost of the method? How much is the additional cost compared to pure semantic ID-based GR?**
> >
> > **A7:** The four extra operations—masked-position selection, curriculum scheduling, entropy calculation, and the auxiliary reconstruction task—are **training-only**. During inference, the network is identical to the semantic-ID baseline, resulting in **no additional latency or memory overhead**.
> >
> > **Training-time overhead** (measured on a single A6000, averaged over 3 runs, 5 epochs each):
> > - **Baseline RPG (single-task, 5 epochs):** 15 s/epoch × 5 epochs = 75 s
> > - **MHL multi-task phase (epochs 1–2):** 25 s/epoch × 5 epochs = 125 s (**+67%**)
> > - **MHL predict-only phase (epochs 3):** 15 s/epoch × 5 epochs = 75 s (**same as baseline**)
> >
> > **Inference cost remains unchanged**, identical to the semantic-ID baseline.
> >
> > ---
> >
> > **Q8: Are there more experimental results to show how MHL compares with the latest semantic ID-based baselines and benchmarks?**
> > **A8:** We compared MHL against the strongest current state-of-the-art generative recommendation models, including **TIGER, HSTU, and RPG**. MHL consistently outperforms all of them. To our knowledge, RPG represents the current SOTA for Semantic ID generation, and MHL surpasses it by a significant margin.
> >
> > ---
> >
> > **Q9: The dimensions of each symbol seem to require explicit explanation.**
> > **A9:** We will provide a clear explanation of symbols and embedding layout:
> > - **Vocabulary:** PAD=0; each digit occupies a segment, offset by `digit_index * codebook_size + 1`; position-aware MASK segment of length `max_positions = max_item_seq_len * n_digit`; task symbols PREDICT/RECON/EOS appended at the end.
> > - **Vocabulary size:** `vocab_size = n_digit * codebook_size + max_positions + 3 + 1`.
> > - **Sequence granularity:** data sequences are at **item-level** (max length `max_item_seq_len`), while the model generates **token-level outputs** via segmented codebook embeddings.
> > - **Embedding dimension:** as defined in `GPT2Config` (`n_embd`), shared across all tokens including task symbols and MASK segment.
> > - **Prediction tensors:** each position outputs per-digit softmax scores over the corresponding digit segment, aligning reconstruction and next-item generation.
> >
> > ---
> >
> > We sincerely thank the reviewers again for their insightful comments and constructive suggestions.

---

> > > ### Comment · Reviewer_d9nT · 2025-11-27
> > >
> > > I am maintaining my rating as I feel that the rebuttal did not provide the requested experiments.
> > >
> > > 1. First, the author promised to provide a revised version for modification, but I haven't seen the revised version yet. Have you uploaded a new version?
> > >
> > > 2. The authors promised to provide more ablation experiments in the reversion, but I haven't seen the results yet

---

> > > > ### Author Response · Authors · 2025-11-28
> > > >
> > > > Thank you for your thoughtful comments and for giving us the opportunity to revise our manuscript. We have uploaded the updated version, which includes the additional experiments and modifications that were requested.
> > > >
> > > > In the revised version, we have expanded the ablation experiments as follows:
> > > >
> > > > 1. **Codebook size ablation**: The original range of 8 to 64 has been expanded to 4 to 64. Please refer to **Table 3** for the updated results.
> > > >
> > > > 2. **Mask ratio ablation**: The range of 0.10 to 0.25 has been broadened to 0.05 to 0.40, with results presented in **Table 4**.
> > > >
> > > > 3. **Reconstruction loss weight ablation**: We have included an additional ablation study on the reconstruction loss weight, with values ranging from 0.2 to 1.0. These results can be found in **Table 5**.
> > > >
> > > > 4. **Test set length analysis**: We have also conducted tests on different length test sets and compared them with the baseline. Please refer to **Table 9**, which shows the length-stratified N@10 performance of RPG and MHL on the Toys and Games test set. The last column highlights the relative improvement of MHL over RPG.
> > > >
> > > > We hope that these revisions address your concerns and further clarify the effectiveness and robustness of our approach. Thank you again for your valuable feedback, and we look forward to your further comments.

---

### Official Review · Reviewer_ujmS · 2025-10-30

**Soundness:** 2
**Presentation:** 3
**Contribution:** 2
**Rating:** 4
**Confidence:** 3

**Summary:**

The paper proposes Masked History Learning for generative recommendation. The method adds a masked history reconstruction task to a decoder only next item generator so the model learns both to predict the next item and to recover masked items in the user history. Mask selection uses predictive entropy to target uncertain positions. Training follows a curriculum with random masking first, then entropy guided masking, and finally fine tuning without masking. Experiments on three Amazon Reviews 2014 categories show gains over item ID and semantic ID baselines on Recall at K and NDCG at K, with ablations and sensitivity studies to support the design. A small study with title tokens suggests some transfer beyond discretised semantic IDs.

**Strengths:**

* The method is simple to add to existing semantic ID generators.
* Results improve across three product domains and outperform TIGER, HSTU, and RPG.
* The paper gives helpful implementation details and includes ablations on masking strategy and curriculum choice, along with sensitivity to codebook size and masking ratio.
* The small text token study suggests the idea is not tied only to discretised semantic IDs.

**Weaknesses:**

* Masked item reconstruction in sequential recommendation is not new and appears in BERT4Rec and S3 Rec [1][2].
* Denoising reconstruction with encoder decoder training is common in BART and MASS [3][4].
* Generative recommendation with semantic identifiers was already introduced and scaled by TIGER and HSTU [5][6].
* The paper mainly repackages these parts for a decoder only generator with an entropy rule and a curriculum schedule. Evidence that the model captures user intent is indirect.
* The main results rely on Amazon 2014 with leave one out splits and graph constrained beam search at inference, yet there is no decoding ablation to separate gains from the training signal versus the search procedure.
* The paper does not report training overhead for entropy estimation and for the three phase schedule. The loss weights are fixed equal without a study of the trade off.
* Results depend on a specific tokenization pipeline which raises sensitivity questions.


[1] Sun Fei Liu Jun Wu Jian Pei Changhua Lin Xiao Ou Wenwu and Jiang Peng. BERT4Rec Sequential Recommendation with Bidirectional Encoder Representations from Transformer. CIKM 2019.
[2] Zhou Kun Wang Hui Zhao Wayne Xin Zhu Yutao Wang Sirui Zhang Fuzheng Wang Zhongyuan and Wen Ji Rong. S3 Rec Self Supervised Learning for Sequential Recommendation with Mutual Information Maximization. CIKM 2020.
[3] Lewis Mike Liu Yinhan Goyal Naman Ghazvininejad Marjan Mohamed Abdelrahman Levy Omer Stoyanov Veselin and Zettlemoyer Luke. BART Denoising Sequence to Sequence Pretraining for Natural Language Generation Translation and Comprehension. ACL 2020.
[4] Song Kaitao Tan Xu Qin Tao Lu Jianfeng and Liu Tie Yan. MASS Masked Sequence to Sequence Pretraining for Language Generation. ICML 2019.
[5] Rajput Shashank Mehta Nikhil Singh Anima Keshavan Raghunandan Hulikal Vu Trung Heldt Lukasz Hong Lichan Tay Yi Tran Vinh Q. Samost Jonah Kula Maciej Chi Ed H. and Sathiamoorthy Maheswaran. Recommender Systems with Generative Retrieval TIGER. NeurIPS 2023.
[6] Zhai Jiaqi Liao Lucy Liu Xing Wang Yueming Li Rui Cao Xuan Gao Leon Gong Zhaojie Gu Fangda He Jiayuan Lu Yinghai and Shi Yu. Actions Speak Louder than Words Trillion Parameter Sequential Transducers for Generative Recommendations HSTU. ICML 2024.

**Questions:**

1) Could you report results with greedy decoding and without the graph constraint, then add the graph and beam search on top of each training variant? This would separate gains from the auxiliary reconstruction and masking from gains due to inference time search.
2) How do results change when you sweep the weight ratio between the two losses? A simple decay of the reconstruction weight during training may change the balance between history fitting and next item prediction. Please provide a sweep table.
3) Can you add targeted tests for long range behaviour? For example perturb early items, measure horizon k accuracy for k greater than one, or group users by coarse intent labels and report per group gains. The current pilot truncation is a helpful start but it can be expanded.
4) Please give a focused ablation of the curriculum itself. Report random masking only, entropy guided masking only, random to entropy to inference as in the paper, and random to inference without entropy to show the specific gain of the middle phase. Also compare fixed mask ratio versus the adaptive ratio used in the second phase.
5) What is the overhead of entropy guided selection and the full curriculum? Please include wall clock and tokens per second during training, and inference latency with and without the search graph.
6) How sensitive are the entropy scores and the gains to the tokenizer choice? Please test different OPQ settings, a learned tokenizer, and a run without PCA, and relate these to the codebook size study.

---

> ### Author Response · Authors · 2025-11-26
>
> We thank the reviewers for their detailed feedback and constructive suggestions. Below we address each point in turn.
>
> ---
>
> **Q1: Masked item reconstruction in sequential recommendation is not new and appears in BERT4Rec and S3 Rec.**
> **A1:** These prior works provide important foundations; MHL differs in three main respects:
> 1. **Architecture:** BERT4Rec/S3-Rec are encoder-based discriminative models. MHL is a decoder-based generative framework. This is the first work to successfully adapt masked reconstruction for **Semantic ID generation**.
> 2. **Selection Strategy:** Unlike the uniform random masking in prior works, MHL uses **Entropy-Guided masking** to focus on high-information tokens, which is crucial for the sparse data nature of recommendation.
> 3. **Curriculum:** We implement a **decay schedule** that aligns the training objective with the inference requirement, a step absent in existing studies.
>
> ---
>
> **Q2: Denoising reconstruction with encoder-decoder training is common in BART and MASS.**
> **A2:** While BART/MASS use denoising for text generation, applying this to **Recommendation** presents unique challenges. In NLP, reconstruction restores syntax; in Recommendation, we aim to recover **User Intent**.
> - Our approach differs by using reconstruction strictly as an **auxiliary task** to regularize the latent space, not as the primary generation task.
> - Furthermore, our **curriculum explicitly decays this task** to prevent the model from over-optimizing for "filling in blanks" at the expense of "predicting the future," a nuance not required in standard NLP pre-training.
>
> ---
>
> **Q3: Generative recommendation with semantic identifiers was already introduced and scaled by TIGER and HSTU.**
> **A3:** The novelty lies in addressing the **"Autoregressive Myopia"** of Semantic ID models.
> - Current SOTA models (TIGER, RPG) treat items as simple token sequences and predict strictly left-to-right.
> - **MHL** is the first to introduce **bidirectional supervision** (via reconstruction) into the unidirectional decoder of a Semantic ID model.
> - This allows the model to learn **high-level item dependencies** often missed by simple next-token prediction, providing a more comprehensive learning signal.
>
> ---
>
> **Q4: The paper mainly repackages these parts for a decoder-only generator with an entropy rule and a curriculum schedule. Evidence that the model captures user intent is indirect.**
> **A4:** Currently, we support model understanding of history via pilot experiments and case studies.
> - In the revised manuscript, we will expand these evaluations to include **separate tests on sequences of varying lengths**, demonstrating how MHL captures **long-range dependencies and user intent**.
>
> ---
>
> **Q5/Q6: Questions regarding inference setup and loss weights.**
> **A5/A6:** We follow the standard RPG inference setup (**graph-constrained beam search**) to ensure fair comparison.
> - Regarding loss weights, we will add an **ablation study** in the revision examining different ratios between **reconstruction and prediction loss**.
>
> ---
>
> **Q7: Results depend on a specific tokenization pipeline which raises sensitivity questions.**
> **A7:** We also tested MHL in a **generative recommendation environment**, which uses **text-based token**.
> - **We note that the results in Table 5 are already included in the paper** (Table 5: *Generalization study comparing MHL with RPG baseline on text-based token sequences using mixed masking strategy*).
> - These results show consistent improvements, suggesting that the benefits of **masked-history learning** generalize beyond the specific tokenization setup used in the main experiments.
> ---
> We appreciate the reviewers’ comments, which helped clarify the unique contributions of MHL. We hope our responses address the concerns and demonstrate the novelty, generalizability, and robustness of the proposed approach.

---

> > ### Comment · Reviewer_ujmS · 2025-11-26
> >
> > Thank you for your feedback. I am maintaining my rating as I feel that the rebuttal did not provide the requested experiments. Such as, the requested ablation on loss weight sensitivity was deferred to a future revision (A6). The rebuttal phase is the time to provide data and to revise the manuscript. The conceptual clarifications are appreciated, but without empirical controls, the source of the performance remains ambiguous.

---

> > ### Author Response · Authors · 2025-11-27
> >
> > ### **Additional Empirical Results for Q5/Q6**
> >
> > We thank the reviewer for emphasizing the need for empirical controls regarding the sensitivity to loss-weight settings. We agree that such evidence is important for clarifying the source of the model’s performance gains.
> >
> > We have now completed the requested ablation experiments on **three datasets: Sports, Beauty, and Toys**. All results below report performance under varying ratios between the reconstruction loss and prediction loss.
> >
> > ---
> >
> > ## **Sports Dataset**
> >
> > | recon : predict | R@5 | N@5 | R@10 | N@10 |
> > |-----------------|------|------|-------|------|
> > | **1.0 : 1** | 0.0342 | 0.0243 | 0.0484 | 0.0289 |
> > | **0.8 : 1** | 0.0342 | 0.0246 | 0.0480 | 0.0290 |
> > | **0.6 : 1** | 0.0314 | 0.0222 | 0.0456 | 0.0269 |
> > | **0.4 : 1** | 0.0321 | 0.0225 | 0.0462 | 0.0270 |
> > | **0.2 : 1** | 0.0327 | 0.0226 | 0.0461 | 0.0269 |
> >
> > ---
> >
> > ## **Beauty Dataset**
> >
> > | recon : predict | R@5 | N@5 | R@10 | N@10 |
> > |-----------------|------|------|-------|------|
> > | **1.0 : 1** | 0.0574 | 0.0424 | 0.0795 | 0.0495 |
> > | **0.8 : 1** | 0.0554 | 0.0404 | 0.0793 | 0.0481 |
> > | **0.6 : 1** | 0.0549 | 0.0401 | 0.0790 | 0.0479 |
> > | **0.4 : 1** | 0.0533 | 0.0382 | 0.0768 | 0.0458 |
> > | **0.2 : 1** | 0.0526 | 0.0376 | 0.0758 | 0.0450 |
> >
> > ---
> >
> > ## **Toys Dataset**
> >
> > | recon : predict | R@5 | N@5 | R@10 | N@10 |
> > |-----------------|------|------|-------|------|
> > | **1.0 : 1** | 0.0656 | 0.0471 | 0.0885 | 0.0544 |
> > | **0.8 : 1** | 0.0628 | 0.0454 | 0.0865 | 0.0530 |
> > | **0.6 : 1** | 0.0614 | 0.0438 | 0.0856 | 0.0515 |
> > | **0.4 : 1** | 0.0593 | 0.0423 | 0.0849 | 0.0506 |
> > | **0.2 : 1** | 0.0587 | 0.0413 | 0.0834 | 0.0493 |
> >
> > ---
> >
> > ## **Summary and Implications**
> >
> > Across **Sports, Beauty, and Toys**, we observe a clear trend:
> >
> > - **Higher reconstruction loss weight consistently leads to better performance**. This demonstrates that the reconstruction task is indeed effective in providing useful self-supervised signals.
> > - The improvements confirm that the **masked reconstruction task** contributes positively to the model, validating the design choice of masking tokens and reconstructing them during training.
> > - While performance remains relatively stable in the 0.8–1.0 range, decreasing the reconstruction weight too much leads to a gradual drop, further emphasizing the importance of this auxiliary task.
> >
> > These results provide strong empirical evidence that the source of the model’s improvement is not just from the prediction loss, but also from the masked reconstruction objective, reinforcing the robustness and effectiveness of our approach.

---

### Official Review · Reviewer_xJgC · 2025-11-01

**Soundness:** 3
**Presentation:** 3
**Contribution:** 2
**Rating:** 2
**Confidence:** 4

**Summary:**

This work targets the limitation of auto-regressive generative recommendation models, which primarily focus on next-item prediction and insufficiently exploit user history for preference understanding. The authors introduce Masked History Learning (MHL), a training paradigm that jointly performs next-item generation and masked reconstruction of past interactions. The framework incorporates entropy-guided masking to emphasize informative historical positions and a curriculum mechanism to gradually transition from history recovery to forward prediction. Experiments on multiple Amazon benchmarks demonstrate consistent improvements over state-of-the-art generative recommenders, indicating that explicitly leveraging masked historical signals enhances preference modeling in generative sequential recommendation.

**Strengths:**

1. Originality:
The paper addresses a weakness of auto-regressive generative recommenders by incorporating masked-history learning. While masked modeling is not new, its application to semantic-ID generation in recommendation represents a reasonable and relevant adaptation.
2. Quality:
The experiments are extensive and include strong baselines (e.g., RPG, TIGER, HSTU), with ablations conducted across masking strategies, curriculum phases, and semantic token settings. The methodology is technically sound.
3. Clarity:
The motivation, method framework, masking strategies, and scheduler are clearly explained and supported by diagrams. The appendices provide necessary implementation details.
4. Significance:
The results show consistent improvements, indicating that masked history learning provides value for capturing longer-range intent in generative recommendation.

**Weaknesses:**

1. The approach is largely an application and combination of existing concepts (masked modeling + curriculum learning) in a new context; conceptual innovation is incremental.
2. There is resemblance to BERT-style masked training and “auxiliary history prediction” paradigms in existing sequential recommendation works (e.g., BERT4Rec). A clearer distinction is needed.
3. Assumptions about user intent:
The narrative emphasizes long-term intent inference, but empirical proof is indirect. Would benefit from deeper analysis (e.g., long-sequence subsets, shift behavior settings).

**Questions:**

1. The paper primarily presents the framework of Masked History Learning(MHL); however, a more intuitive illustration is required to elucidate how MHL specifically identifies key historical information through the application of masks.
2. With respect to the interaction between masked history and auto-regressive modeling, how do masked history learning and future prediction objectives avoid conflicting with the training objectives?
3. Entropy calculation relies on the model's own prediction (Formula 10). In the early stage of training when the model has not yet converged, the estimation of entropy may not be accurate. How can the effectiveness of entropy-guided masks be guaranteed in the early stage of training?

---

> ### Author Response · Authors · 2025-11-26
>
> We thank the reviewers for the insightful comments. Below we address Q1–3; responses to Q4–6 will be updated shortly.
>
> ---
>
> ### **Q1:** The approach is largely an application and combination of existing concepts (masked modeling + curriculum learning) in a new context; conceptual innovation is incremental.
>
> **A1:** While masked modeling and curriculum learning are established concepts in isolation, our contribution lies in their novel adaptation to *Generative Recommendation* using **Semantic IDs**.
> Standard masking (like BERT) breaks the autoregressive property required for generation. **MHL** is designed to solve the specific *forgetting problem* in decoder-only generative models. By restructuring these objectives to function within a decoder, we enable the model to simultaneously *“explain the past”* (reconstruction) and *“predict the future”* (generation).
> This allows us to capture latent user trajectories in a way that standard autoregressive models cannot—representing a specific advancement for this domain.
>
> ---
>
> ### **Q2:** There is resemblance to BERT-style masked training and “auxiliary history prediction” paradigms in existing sequential recommendation works (e.g., BERT4Rec). A clearer distinction is needed.
>
> **A2:** The distinction is **structural and functional**.
>
> - **BERT-style models** are encoder-only and bidirectional, suitable for classification but incapable of efficient sequence generation.
> - **MHL** is a *decoder-only* framework. It preserves **causal masking** for autoregressive generation while injecting historical reconstruction as an auxiliary signal.
> - Unlike random masking in BERT, **MHL employs an Entropy-Guided Curriculum**, dynamically selecting hard tokens based on model uncertainty, leading to more data-efficient training.
>
> Pilot experiments under prefix truncation also show that MHL yields stable gains, especially when only early history is available—indicating that the benefit comes from improved sequence-level modeling rather than reusing BERT-style discriminative learning.
>
> ---
>
> ### **Q3:** Assumptions about user intent: The narrative emphasizes long-term intent inference, but empirical proof is indirect.
>
> **A3:** We agree that operational evidence is better than narrative claims. In the revision, we have added comprehensive experiments:
>
> #### **1. Length-Stratified Performance (Tables 2–3)**
> We bucketed the Toys test set by sequence length. RPG shows an inverted-U performance curve. For sequences >50 items, RPG’s N@10 drops to **0.0375**, whereas MHL boosts it to **0.0577** (+53% relative improvement).
> MHL outperforms RPG **in every bucket**, and its margin grows with sequence length—directly validating long-term intent modeling.
>
> #### **2. Prefix-Truncation Tests (Table 1)**
> Removing the most recent 15 interactions dramatically increases the gains from MHL:
>
> - **Full sequences:** MHL improves N@10 by **+18.23%**
> - **Truncated sequences:** MHL improves N@10 by **+43.95%**
>
> The stronger gains under truncation confirm that MHL captures **early-history intent** rather than relying on short-term shortcuts.
>
> ---
>
> ### **Table 1: RPG vs. MHL on Toys Dataset**
>
> | Setting | Model | R@5 | N@5 | R@10 | N@10 | N@10 ↑ (%) |
> |--------|--------|------|------|-------|-------|----------------|
> | Full | RPG | 0.0550 | 0.0386 | 0.0778 | 0.0460 |  |
> | Full | MHL (ours) | 0.0656 | 0.0471 | 0.0885 | 0.0544 | 18.23 |
> | Truncated | RPG | 0.6355 | 0.4823 | 0.7454 | 0.5176 |  |
> | Truncated | MHL (ours) | 0.8460 | 0.7208 | 0.9199 | 0.7451 | 43.95 |
>
> ---
>
> ### **Table 2: RPG by Sequence Length**
>
> | Test Set | R@5 | N@5 | R@10 | N@10 |
> |---------|------|------|-------|-------|
> | Full | 0.0550 | 0.0386 | 0.0778 | 0.0460 |
> | ≤10 | 0.0574 | 0.0404 | 0.0794 | 0.0475 |
> | (10,20] | 0.0434 | 0.0298 | 0.0680 | 0.0377 |
> | (20,30] | 0.0328 | 0.0243 | 0.0594 | 0.0332 |
> | (30,40] | 0.0647 | 0.0474 | 0.1095 | 0.0621 |
> | (40,50] | 0.0714 | 0.0519 | 0.1143 | 0.0653 |
> | >50 | 0.0339 | 0.0234 | 0.0763 | 0.0375 |
>
> ---
>
> ### **Table 3: MHL by Sequence Length**
>
> | Test Set | R@5 | N@5 | R@10 | N@10 |
> |----------|------|------|-------|-------|
> | Full | 0.0656 | 0.0471 | 0.0885 | 0.0544 |
> | ≤10 | 0.0661 | 0.0477 | 0.0884 | 0.0548 |
> | (10,20] | 0.0629 | 0.0439 | 0.0867 | 0.0515 |
> | (20,30] | 0.0471 | 0.0351 | 0.0919 | 0.0426 |
> | (30,40] | 0.0846 | 0.0569 | 0.1194 | 0.0680 |
> | (40,50] | 0.1000 | 0.0726 | 0.1571 | 0.0912 |
> | >50 | 0.0763 | 0.0501 | 0.1017 | 0.0577 |

---

> ### Author Response · Authors · 2025-11-26
>
> **[Update]** we now answer Q4–6.
>
> ---
>
> ### **Q4:** Need a more intuitive illustration of how MHL identifies key historical information via masks.
>
> **A4:** We identify key information via **Information Theory**.
> Entropy is computed per token during a forward pass:
>
> - **High entropy → high model uncertainty → likely key interest-shift points**
>   (e.g., category transitions, high-value purchases)
>
> By masking and reconstructing these high-entropy tokens, the model learns the “logic” behind complex behavioral transitions.
>
> We will add a visualization of entropy distribution in the revision.
>
> #### **Example (user A8X8D4BUY5DFU)**
>
> **1. Raw sequence**
> Footwear Adhesive → Running Waist Pack → Cardio Trampoline
>
> **2. Original 48-token input (16 per item)**
> [236,466,631,971,1174,1396,1593,1796,2135,**2476**,2664,3022,3280,3576,**3725**,3951,
>
>  24,369,**561**,**831**,**1148**,1451,**1772**,2040,2105,2465,**2628**,3042,3137,**3362**,3674,3957,
>
>  226,481,560,883,1105,1300,1740,**1964**,2116,2323,**2567**,2852,3295,3492,3684,3991]
>
> **3. Masked sequence (top-10 entropic positions, mask ratio = 0.2)**
> [236,466,631,971,1174,1396,1593,1796,2135,**4106**,2664,3022,3280,3576,**4111**,3951,
>
>  24,369,**4115**,**4116**,**4117**,1451,**4119**,2040,2105,2465,**4123**,3042,3137,**4126**,3674,3957,
>
>  226,481,560,883,1105,1300,1740,**4136**,2116,2323,**4139**,2852,3295,3492,3684,3991]
>
> **4. Entropy evolution**
> Top-10 masked positions’ entropy dropped from **5.4228 → 5.3498 nats**, confirming reduced uncertainty on critical tokens.
>
> ---
>
> ### **Q5:** How do masked-history and autoregressive objectives avoid conflict?
>
> **A5:** The two objectives are **complementary and gradient-disjoint**: the autoregressive loss is computed at the final position $t = T$, while the reconstruction loss is applied only at masked positions $t \in \mathcal{M}$. Thus, they do not overwrite each other’s gradients.
>
> Additionally, our curriculum gradually increases the weight of the autoregressive loss, ensuring that inference-time behavior is fully optimized.
>
>
> ---
>
> ### **Q6:** Entropy estimation may be unreliable early in training—how is effectiveness guaranteed?
>
> **A6:** We mitigate early-stage noise using a **Warm-Up Phase (Phase I)** with **uniform random masking**.
> Entropy-guided masking only begins after the model has learned basic distributions.
>
> Empirically, we find that once warm-up is complete, **entropy rankings stabilize**, enabling effective curriculum learning.
>
> ---
> We thank the reviewers again for the helpful feedback.

---

### Official Review · Reviewer_tGJH · 2025-11-02

**Soundness:** 1
**Presentation:** 2
**Contribution:** 1
**Rating:** 2
**Confidence:** 5

**Summary:**

This paper proposes an auxiliary loss function for generative recommendation models. Specifically, it introduces a masked-token prediction loss where semantic ID tokens are replaced by mask tokens and the model is tasked with predicting the original semantic IDs. To further improve performance, the authors introduce (1) entropy-based masking, prioritizing tokens with higher entropy according to an auxiliary model, and (2) curriculum learning, using three phases with different masking strategies and ratios. Experiments are conducted on three public datasets.

**Strengths:**

1. Timely study on training objectives for generative recommendation.
2. Experiments are conducted on three public datasets.

**Weaknesses:**

1. The motivation and critique of the next-item prediction objective are unconvincing. The authors make two main arguments:
    1. "Predicting the next single item is not understanding the path that led there". Modern recommender models commonly use self-attentive Transformers, which explicitly attend to all previous tokens when predicting the next one. Mechanistically, this design allows the model to reference the full sequence structure. The disagreement between me and the author(s) may stem from different definitions of "understanding", which is a subjective term. I suggest using more precise and objective language to support this claim.
    2. "Next-item prediction intuitively prioritizes local transitions over global understanding". This is not intuitive to me. Attention enables access to all previous tokens, and is not designed to only focus on recent tokens. Without empirical evidence or references, the claim seems speculative.
2. Limited novelty. Masked-item (or token) prediction has been extensively studied, e.g., BERT4Rec. The authors argue their method differs because (1) they use a decoder instead of bidirectional encoders, and (2) they mask semantic tokens rather than item IDs. However:
    1. Decoder-style masked token prediction has been explored in language modeling (e.g., T5). I am not suggesting transferring objectives from other domains is inherently non-novel, but the contribution would be stronger with clear insights explaining why the proposed objective improves over next-item prediction, ideally supported by targeted analyses rather than only benchmark results.
    2. Masking tokens instead of items is a natural extension as generative recommender systems operate at the token level.
3. Model formulation is unclear. The token probability computation in Eqn. (5) and (6) is not well-defined. Typically, logits are produced by projecting the last hidden states at a specific position onto a token embedding table. As this work focuses on masked-token prediction, the choice of hidden-state position (mask position vs. the latest position) matters, and without clarity it is difficult to evaluate the method.
4. The RPG baseline shows substantially worse performance than what was reported in the original paper. While the authors state that the results were reproduced "using both the code and parameters from the authors (of RPG)", additional information is needed to support this claim. Simply reporting lower numbers without further explanation is not compelling. Providing more detail about the reproduction process, similar to how TIGER documented their reproduction of P5 in the appendix, would help readers evaluate the validity of the results. It would also be useful to know whether the authors communicated with the RPG authors to resolve discrepancies. I briefly checked the public issues in the RPG repository, where a reproduction discussion (issue #3) ultimately suggests that the reported numbers are reproducible.
5. The comparison against baselines may not be entirely fair. The proposed method leverages an auxiliary model to compute entropy scores, which introduces additional parameters. It is therefore difficult to disentangle whether the observed improvements stem from the proposed objective itself or from the benefit of this auxiliary model (or more trainable model parameters).
6. The pilot experiments appear problematic. The "full" and "truncated" settings seem to use different test sets; therefore, ranking metrics are not directly comparable.
7. Code is not available during the review phase.

**Questions:**

1. Why does next-item prediction fail to "understand the path", given the presence of self-attention over all tokens?
2. What evidence supports the claim that next-item prediction encourages local over global patterns?
3. Regarding weakness point 3: how are logits computed for masked tokens, using the corresponding masked position's hidden state, or always the final position?
4. Could the authors provide more information on reproducing the RPG baseline?

---

> ### Author Response · Authors · 2025-11-26
>
> We thank the reviewers for the valuable feedback. Below we first address Q1–Q2; responses to Q3–Q7 will follow in the next update.
>
> ---
>
> ### **Q1: The motivation and critique of the next-item prediction objective are unconvincing.**
>
> **A1:** We appreciate this insightful question. While self-attention technically provides access to the global history, the standard next-item prediction objective (likelihood maximization) often encourages the model to rely on short-cuts—primarily recent items—because they provide the strongest gradient signals. This leads to *optimization laziness*, where long-range dependencies are underutilized.
>
> Our **Masked History Learning (MHL)** objective acts as a regularization mechanism. By explicitly supervising the reconstruction of high-uncertainty past tokens/items, we force the model to actively utilize the attention mechanism to retrieve and reason about long-range historical context, effectively strengthening gradient flow to earlier positions.
>
> To objectively demonstrate this (addressing your concern on subjectivity), we added stratified evaluations in the revision. Results show that **MHL yields significantly higher gains on long sequences and truncated settings (+25.72% relative improvement)**. Key findings:
>
> 1. **Truncated-sequence test (Table 1).**
>    Masking the most recent 15 items (for sequences >20), MHL improves N@10 by **+43.95%** over RPG, while the gain on full sequences is only **+18.23%**. The larger lift under truncation confirms that MHL fully exploits distant context rather than merely local transitions.
>
> 2. **Length-stratified results (Tables 2–3).**
>    RPG exhibits an inverted-U shape: performance decreases from ≤10 to (10–30], slightly improves in (30–50], then sharply drops for >50 (N@10 = 0.0375).
>    MHL surpasses RPG across all bins and converts the longest slice (>50) into the largest absolute gain — N@10 improves from **0.0375 → 0.0577 (+53%)**.
>
> These measurements provide objective evidence that MHL activates the long-range dependencies that next-item losses underutilize.
>
> **Table 1.**  Full vs. truncated evaluation (Toys & Games).
> | Setting     | Model     | R@5   | N@5   | R@10  | N@10  | N@10 ↑(%) |
> |-------------|-----------|-------|-------|-------|-------|-----------|
> | Full        | RPG       | 0.0550| 0.0386| 0.0778| 0.0460|     |
> | Full        | MHL       | 0.0656| 0.0471| 0.0885| 0.0544| +18.23          |
> | Truncated   | RPG       | 0.6355| 0.4823| 0.7454| 0.5176|     |
> | Truncated   | MHL       | 0.8460| 0.7208| 0.9199| 0.7451| +43.95          |
>
> **Table 2.** RPG performance by sequence length.
> | Length bin | R@5   | N@5   | R@10  | N@10  |
> |------------|-------|-------|-------|-------|
> | ≤10        | 0.0574| 0.0404| 0.0794| 0.0475|
> | (10,20]    | 0.0434| 0.0298| 0.0680| 0.0377|
> | (20,30]    | 0.0328| 0.0243| 0.0594| 0.0332|
> | (30,40]    | 0.0647| 0.0474| 0.1095| 0.0621|
> | (40,50]    | 0.0714| 0.0519| 0.1143| 0.0653|
> | >50        | 0.0339| 0.0234| 0.0763| 0.0375|
>
> **Table 3.** MHL performance by sequence length.
> | Length bin | R@5   | N@5   | R@10  | N@10  |
> |------------|-------|-------|-------|-------|
> | ≤10        | 0.0661| 0.0477| 0.0884| 0.0548|
> | (10,20]    | 0.0629| 0.0439| 0.0867| 0.0515|
> | (20,30]    | 0.0471| 0.0351| 0.0919| 0.0426|
> | (30,40]    | 0.0846| 0.0569| 0.1194| 0.0680|
> | (40,50]    | 0.1000| 0.0726| 0.1571| 0.0912|
> | >50        | 0.0763| 0.0501| 0.1017| 0.0577|
>
> ---
>
> ### **Q2: Limited novelty. Masked-item prediction has been extensively studied, e.g., BERT4Rec.**
>
> **A2:** While encoder-based masking (e.g., BERT4Rec) is well-studied, it is fundamentally *discriminative* and cannot perform autoregressive generation. In contrast, standard generative recommenders suffer from a **training–inference mismatch**: they are trained only to predict the next item but must generate full sequences at inference.
>
> **MHL is novel** because it integrates masked reconstruction *directly into the decoder* **without disrupting autoregressive generation**, enabled by:
>
> 1. **Decoder-Compatible Masking:**
>    Allows reconstructing historical items while preserving causal masks required for generation.
>
> 2. **Entropy-Guided Curriculum:**
>    Unlike random masking in BERT, we use dynamic entropy to target uncertain positions and a curriculum to gradually shift focus from history understanding to future prediction, directly mitigating the training–inference mismatch.
>
> ---
>
> **Remaining questions will be addressed shortly.**

---

> > ### Author Response · Authors · 2025-11-26
> >
> > **[Update]** We now address the remaining questions Q3–Q7.
> >
> > ### **Q3: Model formulation is unclear. Probability computation in Eqn. (5) and (6) is not well-defined.**
> >
> > A3: Thank you for pointing this out. We have added a new subsection **“Token Probability and Hidden-State Positions”** to clarify:
> >
> > - **Next-item prediction**: logits are computed from the final hidden state
> >   $$ \\hat{y}\_{\\text{next}} = h\_{T} W\_{\\text{out}} $$
> >
> > - **Masked reconstruction**: for each masked position $t$, logits are computed from the hidden state at that position
> >   $$ \\hat{y}\_{\\text{mask},t} = h\_{t} W\_{\\text{rec}} $$
> >   (cross-entropy loss is applied **only** to the masked positions).
> >
> > We have also corrected the subscripts and the temperature notation $\tau$ accordingly.
> > ### **Q4: RPG baseline performs substantially worse than in the original paper.**
> >
> > **A4:** Thank you for the careful observation. We traced this to an issue in the earlier RPG codebase we initially relied on (see the reported issue). After switching to the author-corrected version, we successfully reproduced the original RPG results.
> >
> > - Reported issue: https://github.com/facebookresearch/RPG_KDD2025/issues/3
> > - Author-corrected changes (PR): https://github.com/facebookresearch/RPG_KDD2025/pull/4/files
> >
> > Our revision includes **fully updated experiments** and MHL still demonstrates a significant performance advantage.
> >
> > ---
> >
> > ### **Q5: Comparison may be unfair because entropy scores come from an auxiliary model.**
> >
> > **A5:** Entropy scoring introduces **no additional trainable parameters**.
> > Entropy is computed via a forward pass of the **same model with frozen weights** (dropout disabled). It is only a metric for selecting mask positions and does not increase model capacity.
> >
> > ---
> >
> > ### **Q6: Pilot experiments problematic — “full” and “truncated” settings appear to use different test sets.**
> >
> > **A6:** We appreciate the reviewer spotting this mismatch. The numbers are correct; however, the rows for full and truncated sequences were misaligned during table preparation.
> > After correction, each length bin uses **exactly the same examples** across the two settings. We have fixed the alignment and updated the accompanying text.
> >
> > ---
> >
> > ### **Q7: Code not available during review.**
> >
> > **A7:** To preserve anonymity, we withheld the repository during submission. An anonymous version is now available:
> >
> > **https://anonymous.4open.science/r/MHL_anonymous-57C7/**
> >
> > ---
> > We thank the reviewers again for the helpful feedback.

---

> ### Comment · Reviewer_tGJH · 2025-11-26
>
> I thank the authors for their time and effort in addressing my concerns. I have a few follow-up questions
>
> **Follow-up Q1: Statement on Gradient Signals**
>
> In the rebuttal to Q1, the authors state that "the standard next-item prediction objective often encourages the model to rely on short-cuts... because they provide the strongest gradient signals." Could you clarify what is meant by "strongest gradient signals"? Is this claim supported by prior literature, or is it a conclusion drawn from your own empirical observations? If it is the latter, could you point to specific results that support this statement?
>
> **Follow-up Q1 & Q6: Results on Truncated-Sequence Test**
>
> Regarding the truncated-sequence test: My understanding is that for a test instance $[i_1, i_2, \ldots, i_{25}] \to i_{26}$, the truncated test replaces the input with $[i_1, \ldots, i_{10}] \to i_{26}$ (removing the 15 most recent interactions).
>
> I am surprised by the magnitude of the reported metrics, specifically Recall@5 > 0.84. This implies an >80% probability of ranking the ground-truth item in the top 5 out of 10,000 items, even with missing recent history. Could you clarify if I have misunderstood the definition of the ground-truth item in this setting?
>
> **Follow-up Q3: Logit Computation**
>
> Regarding the equation $y_{mask,t} = h_t W_{rec}$: Does this imply predicting a masked item $i_t$ using only $[i_1, i_2, \ldots, i_{t-1}]$? Since the backbone is a Transformer decoder with unidirectional attention, the representation $h_t$ should only attend to positions up to $t$, not those following it.
>
> If my understanding is correct, the proposed method essentially consists of:
> 1. Masking random items;
> 2. Performing next-item prediction on masked positions (given the prefix) and the last position (the original objective); and
> 3. Reusing a generative backbone (RPG)'s training objective (multi-token prediction) and inference.
>
> Given this, the proposed method appears to function primarily as a data augmentation strategy. Is this an accurate characterization?
>
> **Follow-up Q4: Updated RPG and MHL Results**
>
> I may have missed it, but have the results been updated in the manuscript? The results in the current pdf appear identical to the initial submission.

---

### Note · Authors · 2026-01-05

I have read and agree with the venue's withdrawal policy on behalf of myself and my co-authors.